# Characterization of Sensorineural Hearing Loss in Children with Alport Syndrome

**DOI:** 10.3390/life10120360

**Published:** 2020-12-18

**Authors:** Jan Boeckhaus, Nicola Strenzke, Celine Storz, Oliver Gross

**Affiliations:** 1Clinic for Nephrology and Rheumatology, University Medical Center Göttingen, 37075 Göttingen, Germany; jan.boeckhaus@med.uni-goettingen.de (J.B.); celine.storz@web.de (C.S.); 2Clinic for Otorhinolaryngology, University Medical Center Göttingen, 37075 Göttingen, Germany; nicola.strenzke@med.uni-goettingen.de

**Keywords:** hereditary disease, hearing loss, type IV collagen, basement membrane, Alport syndrome

## Abstract

Most adults with Alport syndrome (AS) suffer from progressive sensorineural hearing loss. However, little is known about the early characteristics of hearing loss in children with AS. As a part of the EARLY PRO-TECT Alport trial, this study was the first clinical trial ever to investigate hearing loss in children with AS over a timespan of up to six years Nine of 51 children (18%) had hearing impairment. Audiograms were divided into three age groups: in the 5–9-year-olds, the 4-pure tone average (4PTA) was 8.9 decibel (dB) (*n =* 15) in those with normal hearing and 43.8 dB (*n =* 2, 12%) in those with hearing impairment. Among the 10–13-year-olds, 4PTA was 4.8 dB (healthy, *n =* 12) and 41.4 dB (hearing impaired, *n =* 6.33%). For the 14–20-year-olds, the 4PTA was 7.0 dB (healthy; *n =* 9) and 48.2 dB (hearing impaired, *n =* 3.25%). On average, hearing thresholds of the hearing impaired group increased, especially at frequencies between 1–3 kHz. In conclusion, 18% of children developed hearing loss, with a maximum hearing loss in the audiograms at 1–3 kHz. The percentage of children with hearing impairment increased from 10% at baseline to 18% at end of trial as did the severity of hearing loss.

## 1. Introduction

Alport Syndrome (AS) is a rare genetic disorder of type IV collagen formation leading to progressive renal failure, ocular problems and the development of high-frequency sensorineural deafness [1,2,3,4,5,6,7]. Although approximately 70% of patients with AS suffer from progressive sensorineural hearing loss, little is known about the early development and characteristics of hearing loss in children with AS [2].

Pathogenic variants in the COL4A3, COL4A4 and COL4A5 genes encoding for type IV collagen cause AS [2,3,8]. AS can be inherited in an X-linked (XLAS) form, an autosomal recessive (ARAS) form, or patients with a single heterozygous mutation can present as autosomal dominant AS (ADAS). Pathogenic variants in the COL4A5 gene cause XLAS, which accounts for 80–85% of all patients with AS [9,10]. 15–20% of AS is inherited autosomal and is caused by pathogenic variants in COL4A3 and COL4A4 [1,11]. Individuals with heterozygous AS variants also have an increased risk of end-stage renal failure (ESRF) [12]. AS is diagnosed clinically with the help of patient’s history, physical examination, detailed family history, renal biopsy and genetic testing [13]. 

Type IV collagen is an important component in the structure and function of respective basement membranes in kidney, cochlear, and eyes. Type IV collagen, composed of six different α chains, assembles into three different heterotrimers (α1α1α2, α3α4α5 and α5α5α6), which are tissue-specific [14]. The α1α1α2 heterotrimer is a component of all basement membranes but can be partially replaced by more stable α3α4α5 heterotrimers in mechanically particularly stressed areas in the renal glomerular basement membrane (GBM) in the kidney, cochlea and eyes [15,16]. The α3α4α5 chains form a triple helix structure with a tight twisting of the collagen chains due to the presence of glycine at every third amino acid position. Many pathogenic gene variants in AS are glycine-missense mutations, which lead to kinking of the triple helix structure. Other variants lead to premature chain termination and faster degradation. Patients with splice site or truncating variants or with variants that are located at the 5′ end of the gene have a significantly increased risk of extrarenal involvement. In addition, patients with hearing loss are more likely to develop ESRF early, which makes extrarenal involvement an important prognostic factor [11].

In the glomeruli of the kidney, absence or deficiency of α3α4α5 type IV collagen results in decreased mechanical stability and splitting of the GBM, finally leading to hematuria, proteinuria and to progressive renal failure [16]. Similarly, in the cochlea, the developmental isotype switch between α1α1α2 and α3α4α5 isoforms in basement membranes of the spiral ligament and spiral limbus and underneath the basilar membrane underneath the Organ of Corti does not take place in AS, likely resulting in defects of cochlear homeostasis or micromechanics [17].

Hearing loss due to AS has never been described as congenital, and patients would usually pass newborn hearing screening. It is usually first detected by audiometry in late childhood or early adolescence, presenting with bilateral reduction of sensitivity to mid and high frequencies [18,19,20,21]. The risk of developing hearing loss before age of 30 has been reported as 60% for patients with missense variants and up to 90% for patients with other variants in XLAS [11]. In general, previous studies showed that approximately 70% of the adult patients with AS develop hearing loss over time [2]. There is no specific treatment for delaying hearing loss in AS. Angiotensin-converting enzyme inhibitors (ACEi) are standard off-label therapy for delaying renal failure in patients with AS. Registry data have shown that the progress of renal manifestation can be delayed if treatment with ACEi is started before the glomerular filtration rate (GFR) has dropped below 60 mL/min [22]. 

To clarify whether an even earlier start to therapy is safe and effective, the EARLY PRO-TECT Alport trial (NCT01485978) was initiated in 2012 [23]. The trial was the first randomized and placebo-controlled trial to evaluate the safety and efficacy of renin-angiotensin-aldosterone system (RAAS) inhibition in children. Indicating the safety and efficacy of nephroprotective therapy, results of the primary endpoints have been recently published [24]. The aim of the present study is to describe and assess the secondary end-point hearing function characteristics of children with AS who participated in the EARLY PRO-TECT Alport trial.

## 2. Results

### 2.1. Patient Characteristics

The following results are based on 51 patients out of the 66 patients (77%) from the EARLY PRO-TECT Alport trial, from which additional data regarding hearing function were obtained (Table 1). The 2 female and 49 male patients had a mean age of 9.0 ± 4.2 years at baseline, when 18 patients were in AS stage 0, 23 patients were in stage I and 10 patients were in stage II. Mode of inheritance was in 82% X-linked (42/51), in 16% autosomal (8/51) and unknown in one patient (2%). The median albuminuria at baseline was 61 mg albumin/gCrea (IQR 227.4 mg albumin/gCrea). Of these patients, 18 of 51 reported relatives with hearing loss (35%). Within the trial, 35 of 51 patients were openly treated with Ramipril and 16 of 51 patients entered the randomization arm (seven patients received placebo and nine patients received Ramipril).

### 2.2. Clinical Audiological Characteristics

Hearing loss was diagnosed in nine of 51 patients (18%), while 39 of 51 children had a normal hearing ability (77%). The audiological report was not conclusive in three patients (6%). A previously normal-hearing eight-year old did not cooperate during audiometric testing; in a three-year old child, pure tone audiometry was not performed but the report suggests reduced amplitudes of otoacoustic emissions. In a third patient, only an ambiguous Transient Evoked Otoacoustic Emissions (TEOAE) without interpretation was transmitted.

The youngest child with hearing impairment was a seven-year-old girl with XLAS. Mode of inheritance was X-linked in six and autosomal recessive in three of nine children with hearing loss. Hearing was assessed in eight of nine patients using audiograms. In one patient hearing loss was documented in the medical history without severity. Severity of hearing loss was determined by 4-pure tone average (4PTA) of the better ear (normal hearing: 4PTA ≤ 25 dB, mild hearing loss: 26–40 dB, moderate hearing loss: 41–60 dB, severe hearing loss: 61–80 dB, profound hearing loss >80 dB). At baseline, one child had a mild hearing loss and four children had a moderate hearing loss. Three children developed hearing loss during the trial. All three patients developed a mild hearing loss, and one of three patients progressed to a moderate hearing loss within two years.

The genotype–phenotype correlations described for AS are based on the progression of the kidney disease. Half of the six hearing impaired children with XLAS had a variant categorized as severe and the other half had a variant categorized as intermediate (patient 93_03_02, classified as severe, c.2042-2A>G, *p*.(?); patient 93_05_06, classified as intermediate, c.2395G>A, *p*.(Gly799Ser); patient 93_06_08, classified as intermediate, c.2255G>A, *p*.(Gly752Glu); patient 93_07_06, classified as severe, c.4672C>T, *p*.(Gln1558 *); patient 93_10_01, classified as severe, deletion exon 22–28, *p*.(?); patient 93_13_04, classified as intermediate, c.1948G>C, *p*.(Gly650Arg)). Of the three hearing impaired children with ARAS, one child had a homozygous variant categorized as intermediate, one child had two heterozygous variants categorized as severe and one child had two heterozygous variants categorized as intermediate and less-severe (patient 93_06_05, homozygous COL4A3, classified as intermediate, c.2T>C, *p*.(Met1Thr); patient 93_15_01, compound heterozygous COL4A4, classified as intermediate and less-severe, c.2662G>A and c.4922G>A, *p*.(Gly888Arg) and *p*. (Cys1641Tyr); patient 93_16_03, compound heterozygous COL4A3, classified as severe and severe, c.1398del and c.4348C>T, *p*.(Asp466Glufs * 32) and *p*. (Arg1450 *)).

### 2.3. Correlation between Renal Function and Hearing

As all children with AS in the EARLY PRO-TECT Alport trial were at the early stages of renal disease with normal glomerular filtration rate, renal function was assessed by the amount of albuminuria: in 36 children an audiogram and a test for albuminuria were performed simultaneously. In two patients with normal hearing an albumin excretion was not available at the time of the audiogram. Patients with normal hearing (*n =* 28) showed a median albuminuria of 45.8 mg albumin/gCrea, while median albuminuria of children with hearing impairment (*n =* 8) was higher (300 mg albumin/gCrea) (Figure 1). The logarithmic scale of albuminuria differed significantly (*p* ≤ 0.05) between children with hearing loss and children with normal hearing.

At baseline five of the nine children with hearing loss were in stage I of AS (microalbuminuria: 30–300 mg albumin/g creatinine (gCrea)) and the remaining four hearing impaired children were in stage II of AS (proteinuria: >300 mg albumin/gCrea). Hearing loss was not observed in children in stage 0 (microhematuria without microalbuminuria). 

### 2.4. Audiograms

In 38 children (30 patients with normal hearing and eight patients with hearing loss), one or several audiograms were available. The audiograms were divided into three age groups, 5 to 9, 10 to 13 and 14 to 20 years (Figure 2). 30 patients had one audiogram, seven patients had an additional follow-up audiogram and one patient with hearing loss had an audiogram in each age group (three audiograms). The average time between examinations was 40 months. Accordingly, including nine follow-up audiograms, a total of 47 audiograms were analyzed. The mean 4PTA of the 5–9 years old children was 8.9 dB (*n =* 15) for patients with normal hearing and 43.8 dB (*n =* 2) with impaired hearing. For the 10–13 years old children, the mean 4PTA was 4.8 dB for children with normal hearing (*n =* 12) and 41.4 dB for children with impaired hearing (*n =* 6). For the 14–20 year old children, the mean 4PTA was 7.0 dB (normal hearing; *n =* 9) and 48.2 dB (hearing impaired, *n =* 3). Respectively among the 5–9-year olds, the proportion of hearing loss was lower (12%) compared to the 10–13- and 14–20-year olds (33% and 25%). The mean 4PTA of 14–20 year old children was higher in comparison to the 10–3 year old children. The follow-up of five hearing impaired children showed an annual progression of hearing loss between 0.3 dB up to 9.7 dB.

The audiograms showed clear differences between children with and without significant hearing loss (Figure 3). Thresholds of normal-hearing subjects were typically 5–10 dB across all frequencies. Only four of 36 audiograms classified as normal-hearing according to WHO guidelines showed a 4PTA between 16 and 25 dB, of whom two later developed hearing loss, one patient was not followed up and in one patient the threshold increase was attributed to acute middle ear effusion.

The average tone audiograms of children with hearing loss showed a typical broad mid-cochlear dip with a maximum at 1–3 kHz (trough-type). The curve of the 5–9 year old patients has a maximum dip at 1–2 kHz. In the 10–13 year old patients, the dip spread to 1–4 kHz. In the oldest group of the 14–20 year old patients, there was additional high-frequency hearing loss at 6 and 8 kHz (plateau-type). The maximum hearing loss did not exceed 60 dB.

Out of eleven audiograms with a 4PTA > 25 dB, six were classified as symmetrical and five as asymmetrical. From a total of 22 audiograms (eleven per side) from eight children with impaired hearing, the maximum hearing loss was in the mid-frequency range in 64% (14/22), and in 18% (4/22) it was in the high frequency range (Table 2). In 18% the maximum hearing loss was between mid and high frequencies (4/22). The 22 tone audiograms assumed the following configurations: two times a flat-type, one time a descending-type, nine times a trough-type with a depression in the mid-range frequencies (“cookie bite”) and ten times when thresholds were normal for low frequencies and elevated in both the middle and high frequency regions (plateau-type).

### 2.5. Long Term Tracking of Hearing Impairment in Individual Patients

In two patients we were able to analyze the course of hearing impairment over a time-span of 8 years and 4 years: Hearing loss of patient A was found before AS was diagnosed when he was eight years old and was treated with hearing aids. The first audiogram showed a symmetrical broad trough-type with a maximum at 1–2 kHz (Figure 4A). The medical history of the patient indicated regular language development in early childhood, suggesting normal hearing function. Consistent with an impairment of active cochlear amplification, Transient Evoked Otoacoustic Emissions (TEOAE) were not detectable on both sides. Over the years the maximum hearing loss increased in the high frequencies. The 4PTA changed from a minimum of 43.8 dB to a maximum of 50 dB. Overall, hearing loss hardly changed between the ages of 8 and 16 in the frequencies of 0.25 to 2 kHz, but increased to 12.5 dB at 3 kHz, to 20 dB at 4 kHz, 15 dB at 6 kHz and 25 dB at 8 kHz. 

Patient B developed bilateral hearing loss during the trial. At the age of 8 years, the audiogram showed mild threshold elevations, but hearing loss was not significant according to WHO classification and TEOAE were present in the right ear for all frequencies and in the left ear at 2–4 kHz. The perception of monosyllabic words (Göttinger speech test 2) was slightly impaired with 80/90/90% correct discrimination 55/65/80 dB. At the age of 10 years, hearing loss had progressed especially in the right ear and hearing aids were prescribed. In addition to renal symptoms and hearing loss, the patient also suffered from dyslalia and hyperopia. Within four years, hearing loss (4PTA) increased by 35 dB (right ear) and 37.5 dB (left ear). The last audiogram at the age of twelve showed a symmetrical hearing loss with a broad trough-type pattern with a maximum at 2 kHz (Figure 4B).

## 3. Discussion

The EARLY PRO-TECT Alport trial evaluated the safety and efficacy of early therapy with Ramipril in children with AS. This study was the first clinical trial ever that investigated hearing loss in children with AS (as secondary end-point). The proportion of children with hearing loss in our trial was lower (18%) than described in the literature (about 40% of 11-year-old children with hearing loss) [11,21]. In adults, previous studies showed that approximately 70% of patients with AS develop hearing loss over time. Possible reasons for the smaller proportion of hearing loss in our trial could be the early disease stages and the milder variants in our study population. 

Hearing loss in AS is never congenital and described as first appearing in late childhood or early adolescence [18,19,20,21]. In the EARLY PRO-TECT Alport trial, hearing loss was not observed before children reached elementary school age with the youngest child with severe hearing loss being seven years old. The small number of cases enables only a limited description of the progression of hearing loss. In all of our affected patients, hearing loss did not exceed 60 dB, which is consistent with the literature [11]. According to our data, bilateral sensorineural hearing loss is progressive, with hearing loss between 0.3 dB up to 9.7 dB per year.

Based on these data, we would suggest audiometric testing in normal-hearing AS patients every three years. Children with minimal threshold elevations (4PTA between 16 and 25 dB) should be followed closely, as they have a higher risk of progression to manifest hearing loss. Hearing impaired patients should be fitted bilaterally with hearing aids for which routine technical and audiometric check-ups are typically performed at least once per year. Rehabilitation with hearing aids is usually successful and. despite the fact that the 2 kHz region is particularly important for speech perception, deficits in language acquisition are exceptional in AS. 

Our study has several limitations: first, due to limited financial resources in this Government sponsored trial, we only recommended a hearing test including 4PTA at the start and end of the trial and after three years but were not able to include this as part of the official study protocol. This translated to many hearing tests in most of the children, but not to a complete data set. Second, the EARLY PRO-TECT Alport trial included toddlers with a limited ability to perform hearing tests. Finally, the trial included children with AS at very early stages of disease. Therefore, our trial included a number of children, who are likely to develop hearing impairment later during their course of disease and their hearing impairment will develop over time. In conclusion, we expected the number of children with AS and hearing loss to be less in numbers than in a scenario with adult AS-patients.

Hearing loss is greatest in the mid-frequency range with maximal hearing loss centered around 2 kHz. Hearing loss in AS used to be described as symmetrical [25]; however, in our study only six of eleven audiograms were symmetrical. With age, there is often an additional loss in the high-frequency regions, transforming the audiograms from a broad trough-type towards a gradually sloping pattern. The reason for the tonotopic pattern is unknown. The audiograms are clearly distinct from classical noise-induced hearing loss, which usually presents with a sharp notch at 4 kHz. Animal experimental data suggest that AS may lead to an increased vulnerability to noise-induced hearing loss [26]. Noise exposure could thus possibly be one factor in explaining the variability in the hearing phenotype in humans, and it is conceivable that the tonotopic frequency range of exaggerated noise-induced damage differs from normal ears. Further studies are required to address these questions both in animal experiments and in clinical datasets. 

Regarding the mechanism of hearing loss, early human (and mouse) temporal bone studies demonstrate primarily atrophy of the stria vascularis, but also loss of inner and outer hair cells, and sometimes neural degeneration, whereas a later study described damage near the basilar membrane [27,28,29,30]. Our clinical data would be consistent with global defect of cochlear function, either due to altered cochlear micromechanics or due to stria deficiency. Where tested, changes in otoacoustic emissions, speech perception and auditory brainstem responses were as expected for the pure tone audiometry results of the same patients.

Children with impaired hearing had higher amounts of albuminuria, which corresponds with the association between hearing loss and faster loss of kidney function in AS described in the literature [5]. In our trial, hearing impaired children have a higher amount of albuminuria compared to children without hearing loss, but not all children with high amounts of albuminuria showed hearing loss. Possible reasons for this discrepancy could be the severity of the variants causing AS or external factors such as noise exposure.

## 4. Materials and Methods

### 4.1. Patients

The primary endpoints on renal function of the EARLY PRO-TECT Alport trial have been published recently [24]. Briefly, the EARLY PRO-TECT Alport was the first randomized and placebo-controlled trial to evaluate safety and efficacy of the ACEi Ramipril in children with AS in an up to 6 years treatment period between 2012 and 2019. The Ethic Approval Code is 11/6/11. This study is registered with ClinicalTrials.gov, NCT01485978.

Children with definite diagnosis of AS aged between 24 months and 18 years and normal glomerular filtration rate were included in the trial.

Stages of AS were defined as:Stage 0: Microhematuria without microalbuminuriaStage I: Microalbuminuria: 30–300 mg albumin/g creatinine (gCrea)Stage II: Proteinuria: >300 mg albumin/gCrea

Written informed consent was obtained from all legal representatives and from all patients, who were six years old or older. Children were either randomized, treated with Ramipril or placebo, or openly treated with Ramipril. Children needed to be untreated with an ACEi and to be in stages 0 or I of disease to qualify for randomization. ACEi-pretreated children, children on stage II of disease or those for whom legal representatives denied randomization could be openly treated with Ramipril. During the trial, clinical information about the patient and the medical history of the family were collected using a standardized questionnaire. Family history was considered positive when a family member mentioned any symptoms of hearing loss. All collected data were pseudonymized. 

In 51 of 66 patients, additional data regarding hearing function was obtained from medical reports from specialists in pediatric audiology or otolaryngology, including medical history and audiograms. The data regarding hearing function was collected in the context of regular patient care (hearing test every three years recommended). A pre-existing diagnosis reported in the medical history or an audiogram with a 4PTA > 25 dB of the better ear was classified as hearing loss. In 38 of 51 children, one or several audiograms were available. Genetic testing for the underlying Alport variant was performed in all children included in this present study.

### 4.2. Audiograms

The 4PTA of the better ear was used to classify hearing loss according to the WHO criteria (1997) into mild (26–40 dB hearing loss (HL)), moderate (41–60 dB HL), severe (61–80 dB HL) and profound (>81 dB HL). The audiogram curves were divided into flat-type, trough-type and plateau-type configurations [20,21]. The frequencies of the audiograms were divided into low (0.125, 0.25 and 0.5 kHz), middle (1, 2 and 3 kHz), middle-high (2, 3 and 4 kHz) and high (4, 6 and 8 kHz) frequency regions. 4-Pure tone average (4PTA) was calculated as the mean hearing loss of the frequencies 0.5, 1, 2 and 4 kHz. Air conduction thresholds were used for analysis, unless an air-bone gap ≥10 dB in two neighboring frequencies indicated additional conductive hearing loss. In these cases, bone-conduction thresholds were used. Hearing loss was considered symmetrical if the difference in the same frequency between right and left ear was less than 15 dB [21]. To correlate hearing function with kidney function, each audiogram was matched with albuminuria values obtained within six months from the audiogram.

## 5. Conclusions

Our long-term follow up data originating from a clinical trial confirm that inner ear deafness in children is a very important early sign of AS, which can also be considered to be prognostic factor for progressive kidney disease. Patients with Alport syndrome should have audiometric check-ups to ensure adequate early treatment with hearing aids, and check-ups should start just before elementary school age. In a child with hearing loss and hematuria, genetic testing should exclude or diagnose AS as underlying disease. Future studies should place a special focus on the sociocultural burden and pathogenesis of hearing loss in children with AS, which limits the quality of life before the actual kidney problems show up. 

## Figures and Tables

**Figure 1 life-10-00360-f001:**
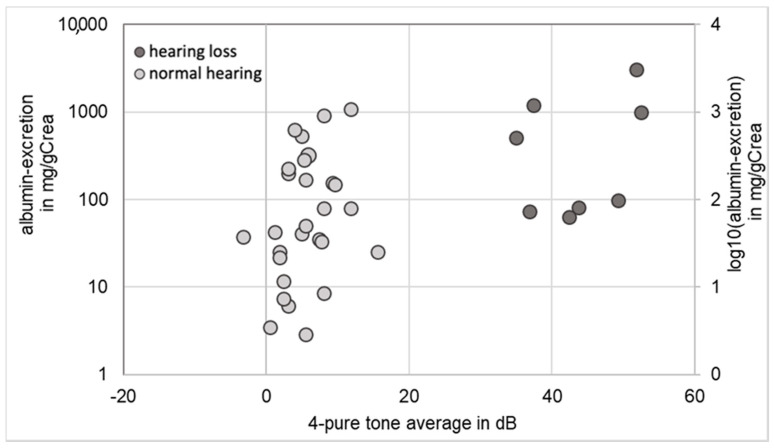
Correlation between hearing ability and albuminuria in eight patients with hearing loss (marked black) and 26 patients with normal hearing ability (*n =* 36).

**Figure 2 life-10-00360-f002:**
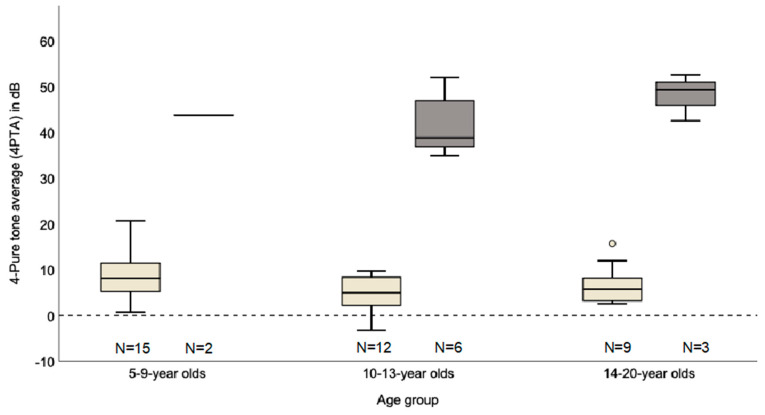
Hearing loss in dB (4–pure tone average (4PTA)) for different age groups; beige normal hearing (4–PTA ≤ 25 dB), gray hearing impairment (4–PTA > 25 dB).

**Figure 3 life-10-00360-f003:**
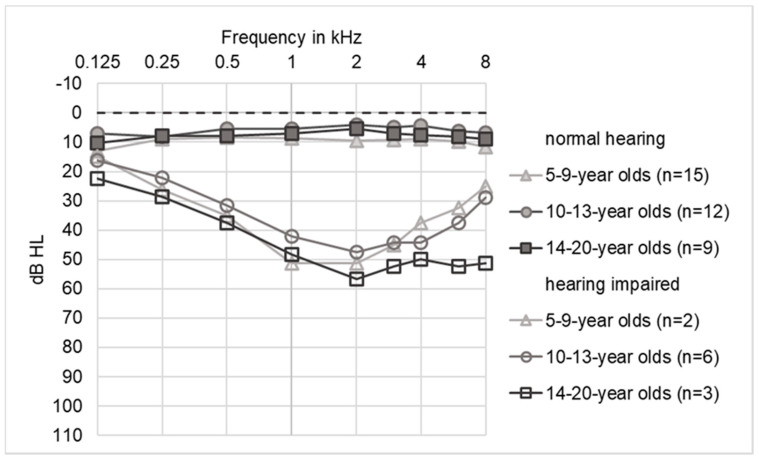
Audiogram for different age groups. Hearing thresholds from both ears were averaged.

**Figure 4 life-10-00360-f004:**
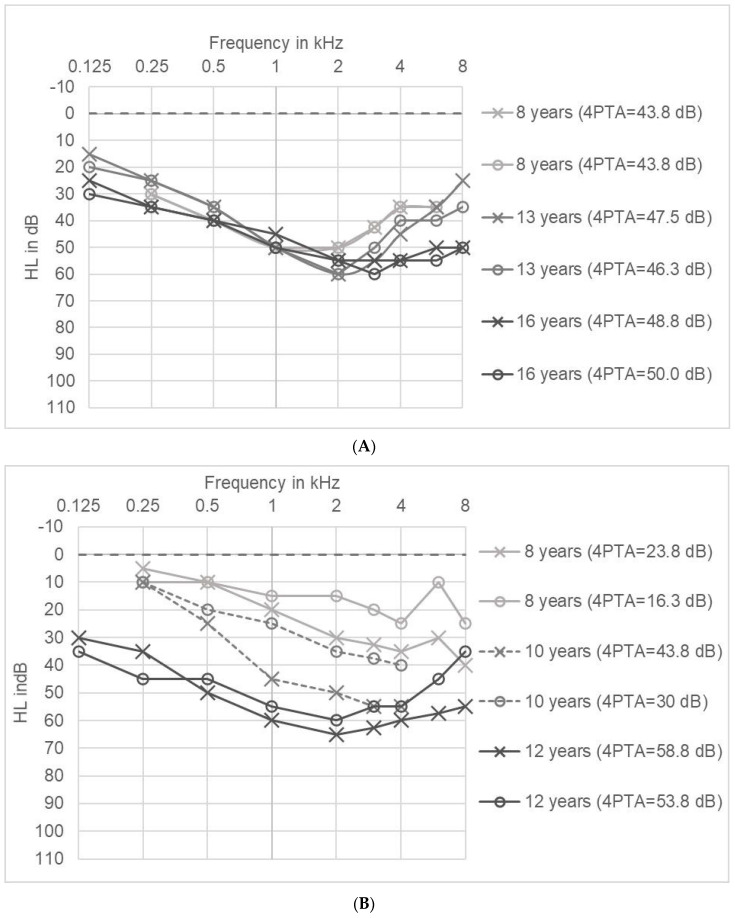
(**A**): Long term tracking of hearing impairment in patient A (air conduction, both ears). Right ear marked with cross (×) and left ear marked with circles (ο). (**B**): Long term tracking of hearing impairment in patient B (both ears; solid lines: air conduction, dashed line: bone conduction). Right ear marked with cross (×) and left ear marked with circles (ο).

**Table 1 life-10-00360-t001:** Baseline characteristics of patients with additional data regarding hearing function in EARLY PRO-TECT Alport trial. IQR = interquartile range; SD = standard deviation.

	Total (*n* = 51)	without Hearing Loss (*n* = 39)	Unknown Hearing Loss (*n* = 3)	with Hearing Loss (*n* = 9)
	*n* (%)	*n* (%)	*n* (%)	*n* (%)
Study group				
Randomised	16 (31%)	14 (36%)	1 (33%)	1 (11%)
Open	35 (67%)	25 (64%)	2 (67%)	8 (89%)
Treatment				
Placebo	7 (14%)	6 (15%)	0 (0%)	1 (11%)
Ramipril	44 (86%)	33 (85%)	3 (100%)	8 (89%)
Mean age in years ± SD				
Baseline	9 ± 4.2	8.9 ± 4.1	3.7 ± 1.2	10.8 ± 4
At last ear examination	11.5 ± 4.3	11.5 ± 4.1	5 ± 2.6	13.7 ± 3.7
Sex				
Male	49 (96%)	38 (97%)	3 (100%)	8 (89%)
Female	2 (4%)	1 (3%)	0 (0%)	1 (11%)
Mode of heritage				
X-linked Alport Syndrome (XLAS)	42 (82%)	34 (87%)	2 (67%)	6 (67%)
Autosomal recessive Alport Syndrome (ARAS)	8 (16%)	5 (13%)	0 (0%)	3 (33%)
*n*/a	1 (2%)	0 (0%)	1 (33%)	0 (0%)
Positive family history	18 (35%)	15 (38%)	2 (67%)	1 (11%)
Median albuminuria at baseline in mg albumin/gCrea (IQR)	61 (227.4)	34.7 (161.1)	82.9 (54,5)	272.5 (505)

**Table 2 life-10-00360-t002:** Classification of maximum hearing loss per ear.

Maximum Hearing Loss Per Ear	5–9 Year Old (*n* = 2)	10–13 Year Old (*n* = 6)	14–20 Year Old (*n* = 3)
low frequencies (0.125, 0.25 and 0.5 kHz)	0	0	0
mid-frequencies (1, 2 and 3 kHz)	4	5	5
middle-high frequencies (2, 3 and 4 kHz)	0	3	1
high frequencies (4, 6 and 8 kHz)	0	4	0

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
