# Peer review of "Characterization of Sensorineural Hearing Loss in Children with Alport Syndrome"

_life, 2020, doi:10.3390/life10120360_

Round 1

Reviewer 1 Report

This study is well written and original.

Some data can be explained (51 patients with data about hearing but only 38 with available audiograms, 13 audiograms can't be found in medical files?; 36/38 patients were represented in figure 1, what about the 2 missing patients?;  one girl with hearing impairment, is she XLAS or ADAS?)

Was it not possible to make audiograms in a prospective way in each patient included in the study? Which criteria are used to make audiograms? Hearing impairment in relatives? Systematic audiograms ? Other? Please detail if possible

Genetics data are available for all the children but nothing is mentioned about it in link with the hearing impairment, it will be interesting even with the small number of children to make (or not) a genotype-phenotype correlations and to comment it.

The audiograms results can be introduced with the explanation for the choice of 4PTA test results.

Please mentions about the limitations of the studies.

Reviewer 2 Report

Please refer to the attached file. 
I cannot paste it here because it is about 11 pages. 

Reviewer 3 Report

Dear Editor,

Thank you for the opportunity to review the manuscript “Characterization of sensorineural hearing loss in children with Alport syndrome”. This study aims to investigate hearing loss in children with AS. Alport spectrum disorders are genetically heterogenic disorders caused by COL4A3, COL4A4 or COL4A5 mutations and therefore diagnostic process of AS spectrum disorders can be challenging. Genetic testing should be performed for an accurate diagnosis.

The author stated that Mode of inheritance was X-linked in six of nine children and autosomal in three of nine children. It is important to accentuate whether these three patients were heterozygous or homozygous carriers of the mutation. It would be very interesting and the paper would gain much more value if the author compared the results of genetic testing with the findings of audiograms and correlated everything together because types of mutation correlate with severity of phenotype.

Round 2

Reviewer 1 Report

Thank you for your modifications.

Author Response

Thank you for the review and your help to improve the manuscript.

Reviewer 2 Report

The authors have generally addressed comments to satisfaction. 

I only have minor comments. 

Minor comments

1. Table 1.
* albumin/gCrea (IQR) -> IQR should be explained. and if IQR is given, the authors need to state whether this is median. -> i.e. should be clearly stated as median (IQR)
* mean age (years) 11.5±4.3 -> this should be clarified as mean ± SD (standard difference)

2. Table 2.
In their response to reviewer 2, authors stated that the patients were all ARAS.
(as follows)
>>> Autosomal, 8 were they ARAS or ADAS? The phenotypes are extremely different.
>>> Response: They were all ARAS.

In addition, in your response to reviewer 3, it is explained that the patients were all ARAS (that the patients had either homozygous or compound heterozygous mutations). 

And yet how come does table 1 still contain ARAS and ADAS? 

Author Response

Thank you for your review and help to improve the manuscript.

Your minor points:

  1. Table 1.

* albumin/gCrea (IQR) -> IQR should be explained. and if IQR is given, the authors need to state whether this is median. -> i.e. should be clearly stated as median (IQR)

* mean age (years) 11.5±4.3 -> this should be clarified as mean ± SD (standard difference)

Response: Thank you for this point. We now added “IQR = interquartile range; SD = standard deviation” to the legend of table 1.

  1. Table 2.

In their response to reviewer 2, authors stated that the patients were all ARAS.

(as follows)

>>> Autosomal, 8 were they ARAS or ADAS? The phenotypes are extremely different.

>>> Response: They were all ARAS.

In addition, in your response to reviewer 3, it is explained that the patients were all ARAS (that the patients had either homozygous or compound heterozygous mutations).

And yet how come does table 1 still contain ARAS and ADAS?

Response: Sorry for this mistake, we removed the “ADAS” from the table 1.

Reviewer 3 Report

Dear author

When I asked for genotype-phenotype correlation, I meat to add mutations detected by genetic testing, please add in a table column with exact mutation.

You stated in response: The heterozygous child had one variant categorized as less-severe and one variant categorized as intermediate. The other heterozygous child had two different variant categorized as intermediate.   Child with two heterozygous mutation is called compound heterozygote and clinically it can manifest as homozygous autosomal recessive Alport.

Author Response

Dear author

When I asked for genotype-phenotype correlation, I meat to add mutations detected by genetic testing, please add in a table column with exact mutation.

You stated in response: The heterozygous child had one variant categorized as less-severe and one variant categorized as intermediate. The other heterozygous child had two different variant categorized as intermediate.   Child with two heterozygous mutation is called compound heterozygote and clinically it can manifest as homozygous autosomal recessive Alport.

Response: We now added the exact mutation to the last paragraph of section 2.2 (lines 112 to 120).